# Perceived coercion, perceived pressures and procedural justice arising from global lockdowns during the COVID-19 pandemic: A scoping review

Veronica Ranieri [1,2]*, Sunjeev K. Kamboj [1], Sarah J. L. Edwards[2]

**1** Research Department of Clinical, Educational and Health Psychology, University College London, London, United Kingdom, **2** Department of Science, Technology, Engineering and Public Policy (STEaPP), University College London, London, United Kingdom

* v.ranieri@ucl.ac.uk

**Data Availability Statement:** All available data have been included in the paper in the form of a summary table.

## Abstract

This aim of this scoping review is to map what is known about perceived coercion, perceived pressures and procedural justice within the context of the general population's experience of 'lockdowns' imposed by governments worldwide in response to the increased transmission of COVID-19. Arksey & O'Malley's (2005) framework for conducting scoping reviews was chosen. A sensitive search strategy was devised and conducted using PubMed, Scopus, and Web of Science using the following search terms: (adherence OR acceptance OR agreement OR trust OR distrust OR compliance OR willing*) OR (perceived coerc* OR percept* coerc* OR pressure OR force OR influence OR control OR threat OR justice) AND (lockdown) AND (COVID OR SARS-CoV-2 OR COVID-19). The database search initially produced 41,628 articles to screen. A total of 40 articles were included in this review and the following five themes were identified from the studies: perceived acceptability and willingness to adhere to lockdown; perceived control during lockdown; perceived pressures arising from lockdown; perceived threat of sanction from others and the procedural (in)justice of lockdown. Our synthesis suggests that i) individuals experienced an initial willingness and tolerance of lockdown that lessened over time as perceptions of personal control decreased; ii) that social influences may pressure individuals to follow or break lockdown rules; and iii) that justifiability and proportionality together with individuals' perceptions of harm from COVID-19 may impact the extent to which individuals adhere to lockdown. Furthermore, the review found an absence of information regarding specific individual characteristics and circumstances that increase the likelihood of experiencing perceived coercion and its related constructs and highlights a need for a better understanding of the cultural and socioeconomic factors affecting perceptions of, and adherence to, lockdown.

**Funding:** SKK's work was partly supported by a grant from the Sir Bobby Charlton Foundation. The funders had no role in study design, data collection and analysis, decision to publish, or preparation of the manuscript. No additional external funding was received for this study.

**Competing interests:** The authors have declared that no competing interests exist.

## Introduction

Perceived coercion, a term borrowed from the mental health literature in relation to mental health hospital admissions, describes the extent to which an individual believes they have choice, autonomy and control over their admission [1]. Within a mental health hospital context, such perceptions are observed when individuals view their detention as pressured or forced, or when they feel excluded from the decision-making process prior to their admission, believe that they were not given a voice, and express that the reasoning behind their admission was unjustified or unfair [2]. Understanding whether such perceptions are present is important as they have been linked to poorer treatment outcomes, poor therapeutic alliance [3, 4], dissatisfaction with mental health services [5], diminished out-patient treatment adherence as an out-patient [6] and disengagement from mental health services [2].

In the context of mental health practice, detention may be justified to protect the individual and others from harm and professionals are ethically bound to review and limit the use of restrictive practices (i.e. involuntary detention, seclusion) that may give rise to perceived or actual coercion, to ensure that any measures that limit liberty or autonomy are lawful and continue to be morally justified. Similar clinical and ethical considerations apply to physical public health care crises. However, in 2020, in response to the escalating rates of transmission of COVID-19, many individuals experienced restrictions on freedom of movement (such as lockdown and quarantine) imposed by governments worldwide. When managing a public health crisis, Mill's Harm Principle can be applied to restrict liberty for the protection of others from harm or in the best interest of the public [7]. Although both types of restrictions are imposed unto an individual or group of individuals in relation to the presence of an illness, public health restrictive measures (i.e. lockdown) pertaining to COVID-19 presented an additional ethical challenge as severity of symptoms can vary throughout the population, with some experiencing asymptomatic transmission while others experienced life-altering disease or death. Though we have a relatively well-developed understanding of the impact of such restrictions on infectious disease transmission, we do not yet know about the implications of such restrictions on future adherence to and engagement with public health messaging and psychological wellbeing during public health crises such as the COVID-19 pandemic. It is also less clear whether there are clinical and cultural variables that may account for differences between individuals' and countries' responses to such restrictions. This is important as scientists have forewarned of the possibility of future epidemics that may require the use of similar or more severe restrictions [8].

In light of the above, a scoping review was conducted to broadly map what is known about perceived coercion and the attitudes of the general population towards lockdowns imposed by governments worldwide in response to the spread of COVID-19. The purpose of this review is to inform both our understanding of and public health policies on the factors that contribute to greater perceived coercion, with a view to comprehending how such factors may impact on psychological wellbeing and other affiliated factors.

## Materials and methods

The aim of this scoping review was twofold: 1) to map out what is known on perceived coercion and/or related constructs, in relation to the COVID-19 lockdown globally, and 2) to identify and emphasise gaps in knowledge within the topic which may motivate future research. Our primary questions were the following: 1) What is known, in the literature, about perceived coercion and its components in relation to the COVID-19 lockdown? 2) How did individuals across the world perceive the COVID-19 lockdown or stay-at-home restrictions in their

individual countries? 3) What factors influence individuals' perceptions of coercion in relation to the COVID-19 lockdown?

Though debates on the use of coercion in preventing the propagation of infectious disease have taken place historically, the psychological concept of perceived coercion has not formally been applied to the context of a pandemic before. Thus, a scoping review, rather than a systematic review, was deemed appropriate for synthesizing and widely mapping areas relating to this concept within the literature. By applying a scoping review methodology, a broad spectrum of studies with varying research methodologies were included, ranging from editorials to systematic reviews, inclusive of both qualitative and quantitative research designs. Arksey & O'Malley's framework for conducting scoping reviews provided a skeleton for this review [9]. Using this framework, the review began by determining a research question and search terms in order to locate appropriate studies from the literature. Selected studies were then reviewed, extracted, and reported within the results section below.

## Search strategy

A sensitive search strategy was conducted using PubMed, Scopus, and Web of Science. Search terms were: (adherence OR acceptance OR agreement OR trust OR distrust OR compliance OR willing*) OR (perceived coerc* OR percept* coerc* OR pressure OR force OR influence OR control OR threat OR justice) AND (lockdown) AND (COVID OR SARS-CoV-2 OR COVID-19). Search terms such as acceptance, agreement and willingness were included as the presence of these may imply lower perceived coercion whilst the absence is likely to signify the presence of perceived coercion. Other search terms were also tested but excluded because of the limited relevance of the resulting studies. Articles were included if they pertained to COVID-19-related lockdowns (i.e. where individuals were legally mandated to stay at home) and assessed attitudes and behaviours relevant to perceived coercion (i.e. acceptance, agreement, trust, compliance or willingness) or a key component of the main measure of perceived coercion in healthcare settings (the MacArthur Admission Experience Survey (AES), i.e. perceived pressures, coercion, force, influence, control, threat, (in)justice) [1]. Articles were excluded if they did not refer to the COVID-19 pandemic or lockdown, if they did not pertain to community samples of adult participants (≥18 years) or if they pertained to participant groups outside of the remit of our review (e.g. surgery or asthma patients etc). Examples of excluded search terms were: (adherence OR acceptance OR agreement OR trust OR distrust OR compliance OR willing*) OR (perceived coerc* OR percept* coerc* OR pressure OR force OR influence OR control OR threat OR justice) AND (quarantine OR lockdown OR isolation), as many of the search results were non-specific to the COVID-19 pandemic; and (perceived coerc* OR percept* coerc* OR pressure OR force OR influence OR control OR threat OR justice) AND (COVID-19 or coronavirus), as most results were in relation to mental health admissions.

The search was completed between April-May, 2022. All titles, abstracts, and full-text articles were screened by the first author (VR). The other members of the research team (S.E. and S.K.K.) independently screened 12% of all titles and abstracts (n = 5000), and remaining full texts to ensure that these met the inclusion criteria. Discussion regarding the included and excluded articles between the three researchers also took place to ensure that only relevant articles were included in the review.

## Data extraction

Extracted details included article authors, country in which the research was performed, year of publication, journal title, article type (e.g., editorial/commentary or research), sample

population, study design, and key findings. For a copy of this, please see Table 1. Prevalent similarities or differences found across the literature were grouped into themes. Each theme was categorized by VR and reviewed by all authors. A description of these is presented below.

## Results

The database search initially produced 41,628 articles to screen. Duplicates were identified and eliminated. After applying the inclusion and exclusion criteria at each stage of screening, the majority, 41, 378 articles, were deemed ineligible. The remaining 251 articles were then full-text screened to assess whether these focused on as aspect of perceived coercion during the COVID-19 lockdown in the general population. A total of 40 articles were deemed eligible and included in the review. Please see Fig 1 below for a PRISMA flow chart diagram of the screening process.

### Types of literature

The majority of articles originated from European countries (52.5%, n = 21). The remaining articles originated from Asia (20%, n = 8), the Americas (7.5%, n = 3), Australasia/Oceania (10%, n = 4), Africa (5%, n = 2) and the Middle East (2.5%, n = 1). One further study was conducted internationally and included data from 79 countries. Most articles reported novel findings from primary data (80%, n = 32). The remaining articles consisted of five commentaries (13%), one systematic literature review (3%), one letter (3%), and one policy document (3%). Out of all reviewed articles, 70% were quantitative (n = 28), 5% were qualitative (n = 2), and ~3% (n = 1) used mixed methods. Further information on the included articles is outlined in Table 1.

### Identified themes

Five themes were identified from the studies: perceived acceptability and willingness to adhere to lockdown; perceived control during lockdown; perceived pressures arising from lockdown; perceived threat of sanction from others and procedural (in)justice of lockdown, as presented below.

### Theme 1: Perceived acceptability and willingness to adhere to lockdown

The available studies examining individuals' willingness to comply with lockdown reported that participants based in high-income countries such as Saudi Arabia and some European countries generally expressed a willingness to restrict their right to freedom of movement for the protection and health of others [10–13]. Such willingness decreased as individuals experienced frustration and anger over their continued restrictive circumstances, yet increased again when rates of COVID-19 and perceived risk of contracting SARS-CoV-2 rose with subsequent surges as seen in one longitudinal study based in South Africa [14]. A further study, undertaken in Germany, revealed that when presented with differing potential scenarios for lockdown, acceptance of restrictive measures was greatest for the strictest short-term lockdown scenario (e.g. only being allowed to leave the home with official consent and severe penalties for violations) and lowest for lengthier though less restrictive lockdown scenarios (e.g. where citizens could leave their home at certain times, with no potential severe sanctions outside those times [15].

As expected, individuals from North America with right-wing political leanings were less willing to comply with lockdown restrictions [16, 17]. Moreover, willingness to follow restrictions, measured by the absence of oppositional attitudes to lockdown and compliance with

**Table 1. Publication details of all articles included in the scoping review after full-text screening.**

| No | Authors | Year | Title | Country | Journal | Design of study | Measure of coercion | Key Findings |
|----|---------|------|-------|---------|---------|-----------------|---------------------|--------------|
| 1 | G. Alkhaldi, G. S. Aljuraiban, S. Alhurishi, R. De Souza, K. Lamahewa, R. Lau, et al. | 2021 | Perceptions towards COVID-19 and adoption of preventive measures among the public in Saudi Arabia: a cross sectional study | Saudi Arabia | BMC Public Health | Cross-sectional survey | Non-standardised measured designed by the researchers examining perceptions, attitudes COVID-19 and its prevention measures; and willingness to self-isolate | Most (82%) participants willing to self-isolate. Higher income households had higher odds of ability & willingness to self-isolate. |
| 2 | N. Aoki | 2021 | Stay-at-Home Request or Order? A Study of the Regulation of Individual Behavior during a Pandemic Crisis in Japan | Japan | International Journal of Public Administration | Scenario-based experiment | Vignette experiment designed by the researchers | Adding penalties (threat of imprisonment /fines) increased lockdown compliance. Authors suggest this may be due to financial risk and fear of imprisonment; also shame/ embarrassment. |
| 3 | M. A. Arunachalam and A. Halwai | 2020 | An analysis of the ethics of lockdown in India | India | Asian Bioethics Review | Commentary | - | Lockdown forced those without reliable income, sanitation, transport, and food to stay at home, resulting in deaths (unrelated to COVID-19 infection). The implementation of lockdown was unequal in India, with authorities enforcing stringent measures on the poor and vulnerable whilst wealthier citizens were able, for example, to conduct and attend marriages and other ceremonies without facing serious sanctions. |
| 4 | J. Bernacer, J. García-Manglano, E. Camina and F. Güell | 2021 | Polarization of beliefs as a consequence of the COVID-19 pandemic: The case of Spain | Spain | PLoS One | Longitudinal survey | Non-standardised measured designed by the researchers examining attitudes to authorities, control, and individual vs group rights | Participants initially disagreed with notions that the authorities were being intrusive and being controlled by others was intolerable when asked at the outbreak of the pandemic. After multiple weeks in lockdown, authorities were perceived as excessively intrusive and individuals' perceptions regarding the intolerability of being controlled increased. As lockdown came to an end, participants agreed more strongly that individual rights were more important than group necessities and that being controlled by others is intolerable. The majority of left-leaning voters agreed that government authorities were intrusive and that it would be intolerable to be controlled by others before the outbreak, however, changed their opinion as infection rates declined. The opposite was true for right-leaning voters, particularly as restrictions eased. |

(Continued)

**Table 1.** (Continued)

| N° | Authors | Year | Title | Country | Journal | Design of study | Measure of coercion | Key Findings |
|---|---|---|---|---|---|---|---|---|
| 5 | N. Bohler-Muller, B. Roberts, S. L. Gordon and Y. D. Davids | 2021 | The 'sacrifice' of human rights during an unprecedented pandemic: Reflections on survey-based evidence | South Africa | South African Journal on Human Rights | Longitudinal survey | Non-standardised measured designed by the researchers examining individuals' experiences, attitudes, and behaviour during the pandemic | The majority (78%) of participants stated they were willing to sacrifice some human rights to help reduce the spread of COVID-19 and protect safety and health of others. Willingness decreased over time in lockdown as frustration and anger increased, and increased again in Winter peak. Willingness to sacrifice was lower in in white adults, and was not linked to gender, education, and age group. Perceived risk/fear of COVID-19 was linked to public support for temporary reduction in civil liberties. There were differences in willingness according to the type of liberty surrendered: approximately half (56%) were willing to surrender their right to religious assembly and freedom to travel. 31% were willing to suspend the right to attend school or protest. Slightly fewer (27%) were willing to forgo their right to work, and less again (27%) for their right to privacy to be impinged upon. Perceiving political leaders as performing well and beliefs that COVID-19 promoted social solidarity, rather than social division. Personal income was not a significant predictor when controlling for gender, age, race, and education. |
| 6 | J. Cameron, B. Williams, R. Ragonnet, B. Marais, J. Trauer and J. Savulescu | 2021 | Ethics of selective restriction of liberty in a pandemic | Australia | Journal of Medical Ethics | Commentary | - | The article argues that liberty-restricting measures such as lockdowns tend to be justified as necessary for harm prevention to others. It further argues that acceptability of a restriction (and ethical principles such as harm prevention, equality and proportionality) should be assessed via an approach that the authors refer to as 'dualist consequentialist'. This approach highlights both that the harm principle does not address what level of liberty restriction is justified for what level of risk of harm, and that the risk of contracting COVID-19, disease burden and cost to individuals is not equal for all. Although restrictions can be viewed as justified if their utility is for the benefit of society, the paper calls for us to assess the costs and benefits of a restriction at both a population and individual level so that some individuals are not disproportionately affected by costs and experience little benefit from these (i.e., one suggestion is to introduce age-selective liberty restrictions) whilst preventing unjustified discrimination. |

*(Continued)*

**Table 1.** (Continued)

| N° | Authors | Year | Title | Country | Journal | Design of study | Measure of coercion | Key Findings |
|---|---|---|---|---|---|---|---|---|
| 7 | M. Ceylan and C. Hayran | 2021 | Message Framing Effects on Individuals' Social Distancing and Helping Behavior During the COVID-19 Pandemic | Turkey & USA | Frontiers in Psychology | Longitudinal survey | Non-standardised measured designed by the researchers examining individuals' responses to various pandemic-related messages | Those with low-medium COVID-19 fear and locus of control are more influenced by prosocial messages rather than self-interest messages. Those with high COVID-19 fear and locus of control are more inclined to adhere to preventive measures. |
| 8 | J. Clinton, J., Cohen, J., Lapinski & Trussler, M. | 2021 | Partisan pandemic: How partisanship and public health concerns affect individuals' social mobility during COVID-19. | USA | Science Advances | Longitudinal survey | Non-standardised measured designed by the researchers examining individuals' willingness to stay at home | Republican voters were less willing to stay at home during pandemic. |
| 9 | M. Constantinou, A. T. Gloster and M. Karekla | 2021 | I won't comply because it is a hoax: Conspiracy beliefs, lockdown compliance, and the importance of psychological flexibility | Cyprus & Greece | Journal of Contextual Behavioral Science | Cross-sectional survey | Measure previously constructed by authors to assess likelihood of adhering to governmental and standardised self-report state and context sensitive measure of psychological flexibility | Belief in conspiracy theories appeared to be a way of coping. The findings suggest that non-compliance may be linked to low psychological flexibility during periods of high distress and that belief in conspiracy theories may provide a sense of meaning and personal control. |
| 10 | J. Farias and R. Pilati | 2021 | Violating social distancing amid the COVID-19 pandemic: Psychological factors to improve compliance | Brazil | Journal of Applied Social Psychology | Cross-sectional survey | Non-standardised measured designed by the researchers examining intentions of noncompliance with social distancing. Perceived behavioral control was measured using an adapted Theory of Planned Behaviour measure. | Stronger perceived behavioural control of violating social distancing was a significant predictor of low compliance. |
| 11 | M. Farina and A. Lavazza | 2020 | Lessons From Italy's and Sweden's Policies in Fighting COVID-19: The Contribution of Biomedical and Social Competences | Italy & Sweden | Frontiers in Public Health | Commentary | - | The authors describe disagreement between experts regarding whether more or less stringent preventive responses to COVID-19 are epistemically justified and scientifically informed. Less stringent responses like in Sweden were based on choice and fewer restrictions on civil liberties but the impact on marginalised groups is not fully known. Discussions about perceptions of restrictive measures should include these groups to protect their ethical and constitutional rights. |
| 12 | R. L. Frounfelker, T. Santavicca, Z. Y. Li, D. Miconi, V. Venkatesh and C. Rousseau | 2021 | COVID-19 Experiences and Social Distancing: Insights From the Theory of Planned Behavior | Canada | American Journal of Health Promotion | Cross-sectional survey | Perceived behavioral control and social norms were measured using an adapted Theory of Planned Behaviour measure. | Perceived control was linked to intention to follow social distancing guidelines, fear of COVID-19 infection and prior social distancing behaviour. |

(Continued)

**Table 1.** (Continued)

| N° | Authors | Year | Title | Country | Journal | Design of study | Measure of coercion | Key Findings |
|---|---|---|---|---|---|---|---|---|
| 13 | M. Gollwitzer, C. Platzer, C. Zwarg and A. S. Goritz | 2021 | Public acceptance of Covid-19 lockdown scenarios | Germany | International Journal of Psychology | Scenario-based experiment | Vignette experiment | Participants' acceptance of restrictive measures was greatest for the strictest short-term lockdown scenario (where citizens were only allowed to leave their homes with official consent and severe penalties for violations) and least for longer lockdown where citizens could leave home when necessary and justified, and with no severe sanctions. |
| 14 | T. O. Gordeeva, O. A. Sychev and Y. I. Semenov [49] | 2020 | Constructive Optimism, Defensive Optimism, and Gender as Predictors of Autonomous Motivation to Follow Stay-at-Home Recommendations during the COVID-19 Pandemic | Russia | Psychology in Russia: State of the Art | Cross-sectional Survey | The Constructive–Defensive Optimism Questionnaire (CODOQ) and Motivation to adhere to recommendations Questionnaire developed by researchers with aim of testing psychometric properties | Both autonomous and controlled motivation were linked to stay-at-home behaviour, with autonomous motivation more strongly correlated with following stay-at-home recommendations. |
| 15 | S. Hills and Y. Eraso | 2021 | Factors associated with non-adherence to social distancing rules during the COVID-19 pandemic: a logistic regression analysis | UK | BMC Public Health | Cross-sectional Survey | Non-standardised measured designed by the researchers examining adherence to social distancing, perceived behavioural control and normative pressure were assessed using an adapted Theory of Planned Behaviour measure | Not adhering to social distancing rules increased if participants felt lower control over leaving the house, lower control over their responsibilities, and lower perception of normative pressure from friends. |
| 16 | J. Jones | 2022 | An Ethnographic Examination of People's Reactions to State-Led COVID-19 Measures in Sierra Leone | Sierra Leone | European Journal of Development Research | Ethnographic study consisting of a cross-sectional survey, interviews, and questionnaires, interviews, participant observations and secondary data analysis of press releases and media reports. | Non-standardised measured designed by the researchers on compliance and non-adherence | Women and children were unable to adhere to lockdown due to nutritional needs. Some passively resisted the lockdown as they perceived it as unnecessary and a were suspicious of COVID-19 funding received from international organisations provided to their government. Others actively resisted the lockdown as they perceived the level of infection as low and questioned how international support would be used by the government to help them. |
| 17 | T. Kamin, N. Perger, L. Debevec and B. Tivadar | 2021 | Alone in a Time of Pandemic: Solo-Living Women Coping With Physical Isolation | Slovenia | Qualitative Health Research | Scenario-based experiment | Vignette experiment | Restrictions were viewed as justified as the virus was perceived as contagious and with potential to cause serious harm, in the absence of a vaccine. On the other hand, limiting personal freedoms /freedom of movement viewed as 'disturbing', particularly to those viewing such restrictions as having a hidden authoritarian agenda. Also present were themes of lack of control over daily life, indifference or acceptance of restrictions. 'Bending' rules by creating bubbles or meeting others outside to counteract isolation. |

*(Continued)*

**Table 1.** (Continued)

| N° | Authors | Year | Title | Country | Journal | Design of study | Measure of coercion | Key Findings |
|---|---|---|---|---|---|---|---|---|
| 18 | D. Krpan and P. Dolan | 2022 | You Must Stay at Home! The Impact of Commands on Behaviors During COVID-19 | UK | Social Psychological and Personality Science | Scenario-based experiment | Vignette experiment | Higher autonomy threat was associated with commanding messages; these lowered intentions to adhere to restrictions. |
| 19 | K. Lachowicz-Tabaczek and M. A. Kozlowska | 2021 | Being others-oriented during the pandemic: Individual differences in the sense of responsibility for collective health as a robust predictor of compliance with the COVID-19 containing measures | Poland | Personality and Individual Differences | Cross-sectional surveys | Measured designed by the researchers on sense of responsibility for collective health with aim of testing psychometric properties | Those who felt concern for others and an obligation to prevent COVID-19 spread were more likely to accept restrictions. |
| 20 | H. J. Lee and B. M. Park | 2021 | Feelings of Entrapment during the COVID-19 Pandemic Based on ACE Star Model: A Concept Analysis | South Korea | Healthcare | Systematic review | - | Authors concluded that "feelings of entrapment" associated with lockdown negatively affect mental health/well-being. |
| 21 | S. Lo Presti, G. Mattavelli, N. Canessa and C. Gianelli | 2021 | Psychological precursors of individual differences in COVID-19 lockdown adherence: Moderated-moderation by personality and moral cognition measures | Italy | Personality and Individual Differences | Cross-sectional survey | Locus of control and moral cognition measured using standardised measures. | Findings indicate that messaging about restrictions must be tailored to two different personalities: 1) promoting greater respect for authority in those who exhibit greater harm-avoidance, and 2) providing clearer and non-contradictory information on the risks for their own health in case of infection for those less trusting in authorities, as threats and sanctions may lead to less compliant outcomes. |
| 22 | S. Lo Presti, G. Mattavelli, N. Canessa and C. Gianelli | 2022 | Risk perception and behaviour during the COVID-19 pandemic: Predicting variables of compliance with lockdown measures | Italy | PLoS One | Cross-sectional survey | Locus of control and moral judgement measured using standardised measures. | Internal locus of control, i.e. the individual perception of being in charge, through voluntary actions, of one's own destiny and life events predicted adherence to restrictions. |
| 23 | A. Maftei and A. C. Holman | 2022 | Beliefs in conspiracy theories, intolerance of uncertainty, and moral disengagement during the coronavirus crisis | Romania | Ethics & Behavior | Cross-sectional survey | Civic moral disengagement and conspiracy beliefs measured using standardised measures. | Lockdown was perceived as more acceptable by those not adopting conspiracy beliefs and in women and those who had a higher level of education. Older participants, those not adopting conspiracy beliefs and with lower moral disengagement were more compliant with the lockdown. |

(Continued)

**Table 1.** (Continued)

| N° | Authors | Year | Title | Country | Journal | Design of study | Measure of coercion | Key Findings |
|---|---|---|---|---|---|---|---|---|
| 24 | K. D. Magnus | 2021 | Commentary: Some Social, Psychological, and Political Factors That Undermine Compliance With COVID-19 Public Health Measures | Germany | International Journal of Public Health | Commentary | - | Compliance or non-compliance with restrictions may be influenced by a desire/pressure to be a part of the 'in-group'. |
| 25 | G. Marinthe, G. Brown, T. Jaubert and P. Chekroun | 2022 | Do it for others! The role of family and national group social belongingness in engaging with COVID-19 preventive health behaviors | France | Journal of Experimental Social Psychology | Mixture of cross-sectional surveys and one longitudinal survey | Intentions to Comply with preventive behaviours measured using standardise measure | Belongingness to social groups predicts compliance with preventive measures. Those close to their families were more intent on complying both to protect themselves and close relatives and vulnerable people from infection. |
| 26 | E. Moser | 2021 | Nozick, the pandemic and fear: A contractualist justification of the covid-19 lockdown | Austria | Global Discourse | Commentary | - | The author presents a 'contractualist' justification for lockdown where government is ethically justified in restricting liberties with the aim of reducing loss of life or serious illness. He notes that there was wide consent for restrictions at the beginning of pandemic that waned over time. The justification for restrictions was based on the potential consequences or fear of negative outcomes that may result from the absence or presence of restrictions. |
| 27 | K. Murphy, H. Williamson, E. Sargeant and M. McCarthy | 2020 | Why people comply with COVID-19 social distancing restrictions: Self-interest or duty? | Australia | Australian and New Zealand Journal of Criminology | Cross-sectional survey | Non-standardised measured designed by the researchers on attitudes to police and government, attitudes regarding enhancing police powers during the pandemic, views regarding the virus, compliance with lockdown restrictions, and the impact that lockdown restrictions had on participants' lives | Compliance with lockdown was linked to duty to support the authorities, and when COVID-19 was perceived as a greater risk to own health. |
| 28 | P. Peretti-Watel, V. Seror, S. Cortaredona, O. Launay, J. Raude, P. Verger, et al. | 2021 | Attitudes about COVID-19 lockdown among general population, France, March 2020 | France | Emerging Infectious Diseases | Letter based on a cross-sectional survey | - | Most individuals were in support of the first lockdown. Low-income participants were less in favour of lockdown and felt such restrictions were disproportionate, coercive and too restrictive, considering risk. |
| 29 | T. Porteny, L. Corlin, J. D. Allen, K. Monahan, A. Acevedo, T. J. Stopka, et al. | 2022 | Associations among political voting preference, high-risk health status, and preventative behaviors for COVID-19 | USA | BMC Public Health | Cross-sectional survey | Non-standardised measured designed by the researchers on attempts to socially distance and on willingness/ability to self-quarantine | Those who reported a preference for Trump were significantly less likely to have tried to socially-distance and to be able or willing to self-quarantine, irrespective of having a high-risk health condition. |

(*Continued*)

**Table 1.** (Continued)

| N° | Authors | Year | Title | Country | Journal | Design of study | Measure of coercion | Key Findings |
|---|---|---|---|---|---|---|---|---|
| 30 | A. Roblain, J. Gale, S. Abboud, C. Arnal, T. Bornand, M. Hanioti, et al. | 2022 | Social control and solidarity during the COVID-19 pandemic: The direct and indirect effects of causal attribution of insufficient compliance through perceived anomie | Belgium | Journal of Community & Applied Social Psychology | Cross-sectional survey | Attitudes towards social control, Solidarity behaviours, disintegration, disregulation measured using standardised measures | Participants who linked the spread of COVID-19 to insufficient compliance with restrictive measures tended to favour greater social control. Perceiving that political authorities were both illegitimate and ineffective was linked with greater social solidarity and less favourable attitudes to social control. |
| 31 | T. Schnell, D. Spitzenstatter and H. Krampe | 2021 | Compliance with COVID-19 public health guidelines: an attitude-behaviour gap bridged by personal concern and distance to conspiracy ideation | Germany & Austria | Psychology & Health | Exploratory longitudinal survey | Self-control measured using a standardised measure | Fear of infection and an external locus of control predicted agreement with restrictions. Opposition to restrictions was low on average. Opposition was greater in those who held more conspiracy beliefs. It was lower in those at greater risk of or more fearful of infection. |
| 32 | L. E. Smith, R. Amlôt, H. Lambert, I. Oliver, C. Robin, L. Yardley, et al. | 2020 | Factors associated with adherence to self-isolation and lockdown measures in the UK: a cross-sectional survey | UK | Public Health | Cross-sectional survey | Non-standardised measured designed by the researchers examining perceived effectiveness of government measures, social pressure, perceived legal consequences of not following government measures and social norms | Poorer adherence to lockdown was linked to lower perceived pressure from friends and family to follow government measures and lower perceived social norms. It was linked to decreased perceived effectiveness of restrictions, illness severity, likelihood of spreading COVID-19 and perceived legal consequences of not following restrictions. Poorer compliance was also linked to fears of losing touch with friends and relatives if followed restrictions, greater general health; believing that you have had or currently have COVID-19; and increased perceived financial cost. |
| 33 | A. Sobkow, T. Zaleskiewicz, D. Petrova, R. Garcia-Retamero and J. Traczyk | 2020 | Worry, Risk Perception, and Controllability Predict Intentions Toward COVID-19 Preventive Behaviors | Poland | Frontiers in Psychology | Cross-sectional survey | Non-standardised measured designed by the researchers examining controllability and intentions towards preventative behaviours | Intent to adhere to restrictions was linked to higher perceived risk and feeling that one has more control over one's circumstances. Willingness to take preventive measures was higher in females and increased with age. |
| 34 | S. Sumaedi, I. Bakti, T. Rakhmawati, T. Widianti, N. J. Astrini, S. Damayanti, et al. | 2021 | Factors influencing intention to follow the "stay at home" policy during the COVID-19 pandemic | Indonesia | International Journal of Health Governance | Cross-sectional survey | Intention to adhere to lockdown policy, perceived behavioural control and subjective norms measured by tool developed by researchers with aim of testing psychometric properties | Subjective norms (ie. perceived expectation to behave in a certain way) and perceived behavioural control were linked to intent to follow "Stay at Home" policy. |
| 35 | Y. L. Tay, Z. Abdullah, K. Chelladorai, L. L. Low and S. F. Tong | 2021 | Perception of the Movement Control Order during the COVID-19 Pandemic: A Qualitative Study in Malaysia | Malaysia | International Journal of Environ Research and Public Health | Interview-based qualitative study | - | Participants expressed that it was their "responsibility as citizens to comply with the MCO regulations". The authors attributed this in part to Asian collectivistic culture which they stated was influenced by normative pressures. |

*(Continued)*

**Table 1.** (Continued)

| N. | Authors | Year | Title | Country | Journal | Design of study | Measure of coercion | Key Findings |
|---|---|---|---|---|---|---|---|---|
| 36 | E. Trifiletti, S. E. Shamloo, M. Faccini and A. Zaka | 2021 | Psychological predictors of protective behaviours during the Covid-19 pandemic: Theory of planned behaviour and risk perception | Italy | Journal of Community and Applied Social Psychology | Cross-sectional survey | Attitude and intention to lockdown, subjective norms, and perceived behavioural control were assessed using an adapted Theory of Planned Behaviour measure | Participants' perceptions of subjective norms and perceived behavioural control predicted intent to adhere to restrictive measures. |
| 37 | V. van Mulukom, B. Muzzulini, B. T. Rutjens, C. J. van Lissa and M. Farias | 2021 | The psychological impact of threat and lockdowns during the COVID-19 pandemic: exacerbating factors and mitigating actions | International (79 countries) | Translational Behavioral Medicine | Cross-sectional survey | Non-standardised measure designed by the researchers examining threat, sense of control, and institutional trust | Days in lockdown did not predict the extent to which individuals felt in control of their circumstances. Coping style and government actions increased sense of control, whilst avoidant action did not. Depressive and anxiety symptoms were predicted by a low sense of control. |
| 38 | L. Wright, E. Paul, A. Steptoe and D. Fancourt | 2022 | Facilitators and barriers to compliance with COVID-19 guidelines: a structural topic modelling analysis of free-text data from 17,500 UK adults | UK | BMC Public Health | Cross-sectional survey | - | Social responsibility and civic duty acted as motivating factors for adherence to restrictions. Social pressure to break rules from family and friends as well as observations of non-compliance among the general public and members of the government acted as barriers to adherence. |
| 39 | J. S. Wu, X. Font and C. McCamley [50] | 2022 | COVID-19 social distancing compliance mechanisms: UK evidence | UK | Environmental Research | Longitudinal survey | Non-standardised measure designed by the researchers examining moral obligation, moral disengagement and behavioural intention | Participants' intent to adhere to restrictions, altruism and moral obligation decreased whilst moral disengagement increased over time. |
| 40 | S. Zadey, S. Dharmadhikari and P. Mukuntharaj | 2021 | Ethics-driven policy framework for implementation of movement restrictions in pandemics | India | BMJ Global Health | Policy framework | - | In the absence of biomedical, epidemiological or other data (ie. at the onset of an epidemic), decision-making must follow ethical principles pertaining to the transparency of communication, accountability, equity, reciprocity, and the use of least restrictive means. As more information unfolds, decisions must be guided by the principles of preventing harm, justifiability and proportionality. |

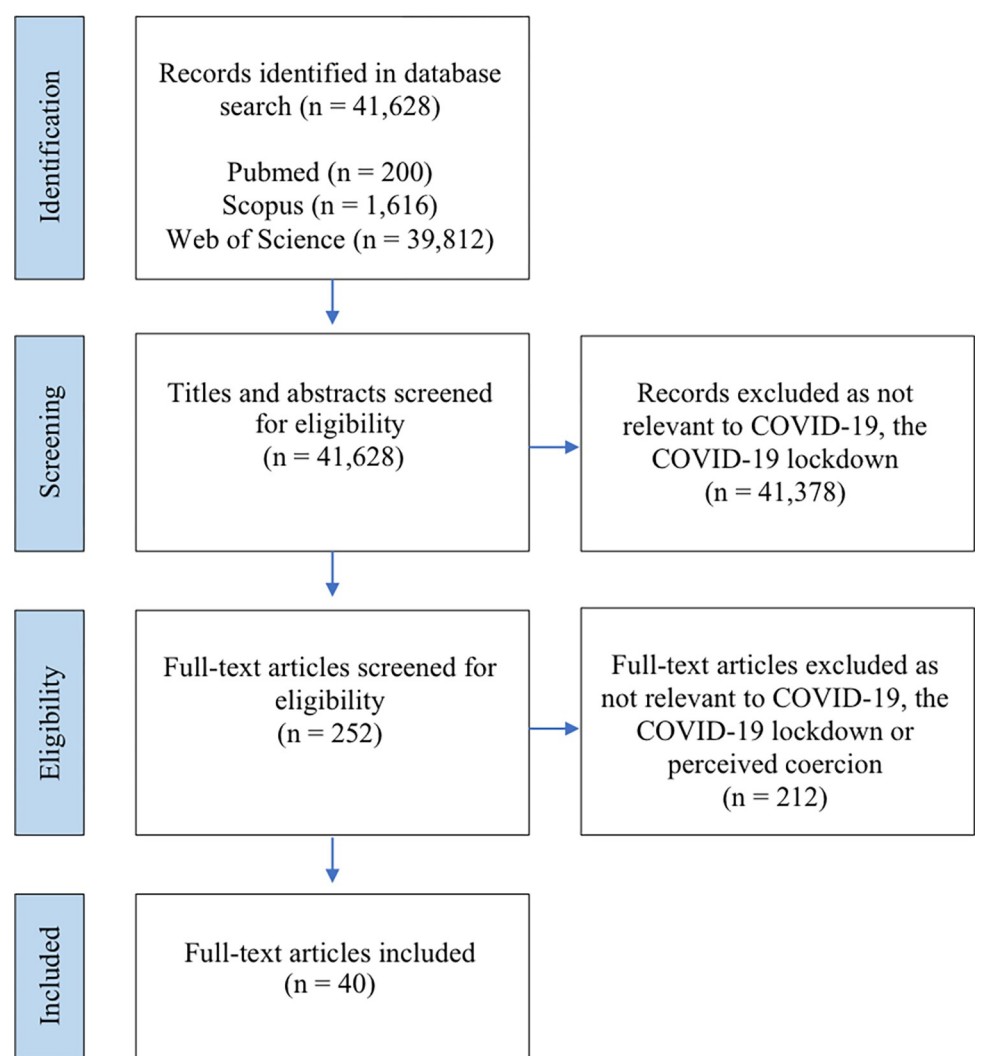

**Fig 1. A PRISMA flow chart diagram of the scoping review screening process.**

such restrictions was positively correlated with beliefs that political leaders were competent and that the aim of such restrictions was to consolidate social solidarity, according to a South African study [14]. Though opposition to lockdowns was generally low across studies, some studies suggested higher rates in participants who expressed scepticism regarding the funding received by their governments from international organisations. Those sceptical about tax-payer funded relief initiatives during the pandemic were also more likely to resist lockdown restrictions, as seen in both low and high-income countries such as Sierra Leone and Slovenia [18, 19]. Opposition to lockdown amongst Europeans from high-income countries tended to be associated with higher COVID-19-related stress (e.g. feelings of intolerability, boredom, anger, fear and pessimism), lower perceived risk of infection, less clarity regarding restrictions and conspiracy beliefs [19–21].

Willingness to live with restrictions also differed according to the which aspect of freedom the restriction curtailed. In the sole South African study, approximately half of participants stated that they were willing to concede their right to religious assembly and freedom to travel, whilst a third were willing to suspend the right to attend school or protest [14]. However, even

temporary restrictions that impacted individuals' ability to travel to work, and their privacy, were viewed as much less acceptable [14]. Socioeconomic characteristics were, in part, linked to perceptions regarding acceptability. Indeed, higher income predicted willingness to sacrifice a broad range of rights apart from the right to work across nations [10, 13, 14]. Similarly, French participants who were financially poorer were less in favour of lockdown, viewing it as coercive and disproportionate to the risk posed by the disease [13]. Of note, nonetheless, are the inconsistent findings within the limited available studies in relation to the impact of other demographic factors on perceived acceptability of lockdowns. For instance, in a South African sample, willingness to adhere to lockdown restrictions was lower in White adults and not linked to gender, level of education or age [14]. Additionally, older age and higher education levels were linked to willingness to accept lockdown restrictions in Eastern European participants [21, 22].

## Theme 2: Perceived control during lockdown

The included studies on perceived control focused on three main areas: 1) the extent to which individuals perceived themselves or others to be in control of their circumstances and their attitudes towards coercive control during lockdown; 2) the impact of perceived control on their psychological wellbeing; and 3) perceived control as a predictor of adherence to restrictions. In a qualitative study, some individuals spoke of not having control over their day-to-day lives whilst others reported feeling indifferent to, or accepting of, restrictions [19]. Nonetheless, included studies highlighted a change in the extent to which people felt in control over their circumstances as lockdowns continued, with individuals' initial sense of tolerance for restrictions and personal perceived control decreasing, and a sense of intrusiveness by authorities increasing as lockdown continued, for instance in Spain [23].

The mental health impact of low perceived control was recorded in two studies. One of these noted that low perceived control predicted depressive and anxious symptomatology in participants spanning 79 countries [24]. Furthermore, feelings of 'entrapment' arising during lockdown and the negative impact of these on individuals' mental health were noted in a prior systematic review [25]. One study noted that belief in conspiracy theories may have acted as a form of coping with distress and satisfied a need for greater control [26].

Both greater perceived control and greater internal locus of control, accompanied by fear of contracting COVID-19 or perceived risk of COVID-19, acted as determinants of willingness to adhere to lockdown in some of the included studies undertaken with European, Turkish and Northern American participants [22, 27–30]. One further study concluded that *external* locus of control was predictive of adherence to lockdown restrictions [20]. Those who did not feel they had the decision-making power to leave their house were less likely to adhere to restrictions [29]. For some of those who lived alone, 'bending' the rules by creating unsanctioned bubbles or meeting outside with others during lockdown was done in an effort to counteract isolation [19]. There is some disagreement within the literature as to whether perceived behavioural control predicted adherence to lockdown, with some studies linking it to intent to adhere to restrictions [31, 32] and others to non-compliance [33].

## Theme 3: Perceived pressures

Social influences and pressure from friends and family was highlighted as an influential factor in how individuals viewed and responded to lockdown regulations. Those close to family members who held favourable views regarding lockdown were positively influenced to comply with regulations to protect themselves, their families and vulnerable others [34]. Conversely, those whose family members did not adhere to lockdown regulations felt lower perceived

pressure to follow such regulations themselves [35]. Pressures to 'belong' or conform to a group identity were also indicative of individuals' attitudes to lockdown, with lower perceptions of normative pressure from friends being predictive of non-compliance in central European countries [29, 34–36]. Individuals who conveyed fears of losing touch with friends and relatives if they followed restrictions tended to oppose to lockdown regulations [37].

Societal norms were also reported to play a role in individuals' perceptions regarding lockdown in a minority of studies. In two qualitative studies examining attitudes to movement control/stay-at-home orders in Malaysia and Indonesia, participants reported that "collectivistic societal norms" pressured them to comply with restrictions and to feel that respecting such regulations was every citizen's duty or responsibility [31, 38]. A sense of social responsibility and civic duty was not exclusive to these South-East Asian studies and was also noted in Australia and some European countries [32, 35, 39]. In one study, individuals who linked the spread of COVID-19 to insufficient compliance with restrictive measures by others tended to favour greater pandemic-related social control [40]. Yet, where members of the general public and government officials were seen to not obey those restrictions, for instance in the UK, the public too felt less pressured and inclined to do so [35].

## Theme 4: Perceived threat from others

Three studies examined how the general population responds to perceiving a threat from others in relation to lockdowns, with inconsistent findings. Two of these studies, both European, indicated that individuals were less likely to respond to commands to stay home if these were perceived as threatening to their autonomy [41, 42]. Another study from Japan, suggested that individuals who perceive a threat of imprisonment or heavy penalty would be more likely to stay at home due to potentially feeling shame associated with such punishments, in addition to fear of financial risk [43]. The extent to which such experiences of shame are culturally determined, is not clear. Both findings suggest that restrictive strategies and their messaging should ideally be tailored to different people. For instance, in Japan, these could either promote respect for authority or provide clearer information on the risks of COVID-19 to those who place less trust in authorities, as threats and sanctions may counterproductively lead to less compliant outcomes [43].

## Theme 5: Procedural Justice of lockdown

The screened literature on procedural justice purports to the ethical justification and fairness of lockdown [44]. According to an Indian policy framework by Zadey, Dharmadhikari & Mukuntharaj (2021), where the extent of harm that a potential pathogen poses is unknown, decision-making and guidelines regarding restrictions of human rights must be clearly communicated, equitable and reciprocal. Such decision-making must uphold the use of least restrictive means and, as more information unfolds, decisions must be guided by the principles of preventing harm, justifiability and proportionality [45]. Other authors in Western cultures focused on such restrictions being justified in light of the risk of negative outcomes to others, particularly in the absence of a vaccine [12, 19, 46]. Nonetheless, examples of discriminative implementation of lockdown and unfair burdens to some of the general population were observed in some studies. For instance, the implementation of lockdown was unequal in India, with authorities adopting stringent measures with the least powerful vulnerable whilst the wealthier were able to conduct and attend marriages and other ceremonies [47]. It also forced those without reliable access to livelihoods, sanitation, transport, and food to stay at home, resulting in deaths that were not related to COVID-19 infection [47].

## Discussion

### Summary of findings

This review provides an initial synthesis of studies relating to the concept of perceived coercion in the context of the COVID-19 pandemic. To reiterate, this concept originally derived from mental health practices that were perceived (by patients) as being coercive and consists of three interrelated constructs: perceived coercion *per se*, perceived control and procedural justice. The reviewed studies suggest that these constructs indeed have relevance to and implications for pandemic-related public health messaging and efforts to promote adherence to restrictive measures (lockdowns). They also highlight differences across geopolitical contexts.

The reviewed studies suggest that participant groups from different countries (with different socioeconomic contexts, cultural norms etc.) were initially accepting of lockdown measures. Acceptance of such measures increased with higher rates of infection and perceived risk of infection. Some of those opposed to lockdown tended to express greater distrust in authorities, held more conspiracy beliefs, viewed the risk of infection as low and the guidance regarding restrictions as unclear. The extent to which individuals felt a sense of control over their circumstances during the pandemic differed across and within studies. Low perceived control was linked to greater depressive and anxious symptomatology [24], and those who felt less in control over their circumstances were less likely to adhere to lockdown [22, 27–30]. Nonetheless, perceived control and tolerance for restrictions lessened over time as a sense of intrusiveness by authorities emerged in one study [23].

In some of the studies, adherence to lockdown was influenced by the views and behaviours of those around an individual. Those with close family members or friends who held favourable views regarding lockdown were influenced to comply with regulations [34], whilst those whose social circles did not adhere to lockdown regulations felt lower perceived pressure to follow such regulations [35]. Messaging too impacted how individuals perceived and responded to lockdown measures. The limited evidence suggests that those who place less trust in authorities may be less likely to respond positively to commanding messages if these are perceived as threatening to their autonomy [41]. Others who are more focused on harm avoidance or who hold greater respect for authority figures may be more likely to stay at home when these perceive a threat that could impose both emotive and financial consequences upon them [43]. However, as the studies pertaining to social norms are few in number, their findings may not be generalisable to other countries. Finally, there is some limited debate regarding the ethicality and fairness of lockdown within the literature, particularly in those with fewer economic means and less reliable access to food, water and sanitation [47]. Some authors argued that decision-making regarding lockdown must adopt the least restrictive means possible until clear information on a pathogen and the risks it poses emerge, whilst others argue that lockdown is justified where there is substantial risk of loss of life [12, 19, 45, 46].

### Implications

As suggested within the review's findings, the adult participants sampled from the general population were more accepting of lockdown where guidance and information regarding risk of illness from a pathogen and resulting restrictions was clear and cohesive and where these were articulated by authorities whom they trusted. Therefore, preparedness for the possibility of future widespread infectious diseases should focus on identifying and incorporating respected members of communities who can clearly convey public health messages and the rationale for restrictive measures. This is important as clear public messaging delivered by trusted and credible figures influences the attitudes both of individuals who receive the messaging and those

around them, with a consequent snowballing influence at a community level [42]. Therefore, messaging from respected, trusted and credible community members may be less likely to be experienced as coercive.

The findings also suggest that those who felt less in control over their circumstances experienced greater anxiety, depression and feelings of entrapment. This has important implications for mental health services as an increase in psychological symptoms may result in greater demand on such services. In countries where psychological distress is more stigmatised, this may result in individuals not having a source of support and containment. One potential solution may be to provide a forum for the general public's voice to feel heard and included when designing public health measures. Another, perhaps more idealistic, option may be to create brief low-intensity psychological intervention referral pathways designed to help individuals with COVID-related anxiety or depression who have less complex psychological presentations (as seen in some Improving Access to Psychological Therapies (IAPT) services in the UK, that provide mental health first aid to healthcare professionals working with patients diagnosed with COVID-19) whilst scaffolding secondary care services [48].

Finally, as highlighted within the included studies, a uniform lockdown can heighten a sense of discrimination among those less privileged and/or historically discriminated against. Under the umbrella of the harm prevention principle, we remain unclear about what level of restriction is justified for what level of risk of harm and, whether the risk of contracting COVID-19, disease burden and cost to individuals is equal for all and proportional to the enforcement of lockdown for all [46]. An assessment of the costs and benefits of lockdown would therefore be warranted to ensure that some individuals are not disproportionately affected by costs and to prevent discrimination [44, 46]. Such an assessment and future policy should aim to provide equitable, rather than equal, support to those at risk of loss of income or access to essential goods.

## Strengths and limitations

In line with the aims of a scoping review, which are to provide a broad overview of the current state of knowledge in a rapidly developing field, this review included various types of literature, ranging from empirical papers with both quantitative and qualitative methodologies, policy frameworks, systematic reviews and commentaries that allowed for the broad mapping of a lesser-known area. Most of the empirical articles employed quantitative online survey methodology. This method ensured that researchers could reach the general population during lockdowns. However, the absence of representative sampling and a consistent measure of perceived coercion, pressures and procedural justice within the general population has serious implications for generalisability. In particular, the samples were biased towards high income countries for whom technology was not a barrier. It also meant that there was less space for participants to speak of their experiences and the meaning they attributed to these in their own voice. The inclusion of commentaries, though biased towards the writer's opinion, also provided some useful philosophical debate on the topic. Missing from this picture is also an account of the grey literature on the topic. Though authors have, more recently, called for the inclusion of a quality assessment in relation to scoping reviews, there is yet to be a comprehensive tool that can uniformly assess a range of methodologies. In the absence of such a tool, we urge caution in interpreting the findings above.

## Supporting information

**S1 File. PRISMA 2020 checklist.**
(DOCX)

## Author Contributions

**Conceptualization:** Veronica Ranieri, Sunjeev K. Kamboj, Sarah J. L. Edwards.

**Data curation:** Veronica Ranieri.

**Formal analysis:** Veronica Ranieri, Sunjeev K. Kamboj, Sarah J. L. Edwards.

**Methodology:** Veronica Ranieri.

**Project administration:** Veronica Ranieri.

**Supervision:** Sunjeev K. Kamboj, Sarah J. L. Edwards.

**Writing – original draft:** Veronica Ranieri.

**Writing – review & editing:** Veronica Ranieri, Sunjeev K. Kamboj, Sarah J. L. Edwards.

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
