## [Decision Letter · Decision Letter 0]

23 Aug 2022

PGPH-D-22-01026

Perceived coercion, perceived pressures and procedural justice arising from global lockdowns during the COVID-19 pandemic: a scoping review

Dear Dr Ranieri,

Thank you for your manuscript 'Perceived coercion, perceived pressures and procedural justice arising from global lockdowns during the COVID-19 pandemic: a scoping review' to PLOS Global Public Health. After careful consideration, we feel that while your manuscript has merit, it does not fully meet PLOS Global Public Health's publication criteria as it currently stands. Therefor, we invite you to submit a revised version of the manuscript that addresses the points raised during the review process. When you revise your manuscript , we would be grateful if attention is paid especially in the analysis to address some of the issues that the reviewers have raised to ensure that your work is relevant to readers of this journal.

As it stands the implications for the global public health community from some of your key findings is not very clear. For example, What implications do you draw for the public health- both policy and practice, from the perceived coercion and justice issues? Are there insights that we can gain that would ensure that we can prepare better (as a community of public health professionals or institutionally) for a future lockdown either local or global? Drawing the implications of your findings clearly for public health ethics and global public health would strengthen your manuscript. I would also suggest a thorough check of the manuscript to ensure that there are no minor errors in grammar and numbers in the PRISMA chart all add up (Reviewer 1's observations)

We look forward to receiving your revised manuscript.

Kind regards,

Mathew Sunil George

Academic Editor

Journal Requirements:

1. Please amend your online detailed Financial Disclosure statement. This is published with the article. It must therefore be completed in full sentences and contain the exact wording you wish to be published.

Please state what role the funders took in the study. If the funders had no role in your study, please state: “The funders had no role in study design, data collection and analysis, decision to publish, or preparation of the manuscript.”

2. Please update your online Competing Interests statement. If you have no competing interests to declare, please state: “The authors have declared that no competing interests exist.”

3. Please provide separate figure files in .tif or .eps format only and remove any figures embedded in your manuscript file. Please also ensure that all files are under our size limit of 10MB.

Additional Editor Comments (if provided):

Reviewers' comments:

Reviewer's Responses to Questions

**Comments to the Author**

1. Does this manuscript meet PLOS Global Public Health’s publication criteria? Is the manuscript technically sound, and do the data support the conclusions? The manuscript must describe methodologically and ethically rigorous research with conclusions that are appropriately drawn based on the data presented.

Reviewer #1: Yes

Reviewer #2: No

2. Has the statistical analysis been performed appropriately and rigorously?

Reviewer #1: Yes

Reviewer #2: N/A

3. Have the authors made all data underlying the findings in their manuscript fully available (please refer to the Data Availability Statement at the start of the manuscript PDF file)?

Reviewer #1: Yes

Reviewer #2: Yes

4. Is the manuscript presented in an intelligible fashion and written in standard English?

Reviewer #1: Yes

Reviewer #2: Yes

5. Review Comments to the Author

Reviewer #1: Perceived coercion, perceived pressures and procedural justice arising from global lockdowns during the COVID-19 pandemic: a scoping review

Abstract

“Arksey & O’Malley’s (2005) framework for conducting scoping reviews” The sentence seems incomplete; can include was used

Whether perceived coercion, pressure, and procedural injustice were reported by medical professionals or by the general public is not clear from the abstract.

Introduction

Few sentences are difficult to understand (highlighted in the manuscript).

The introduction discusses perceived coercion in relation to admissions to mental health facilities. The rationale involves perspective of the general population.

Materials and Methods

Details regarding methods followed while implementing Arksey & O’Malley’s (2005) framework can be provided.

Search Strategy: Other search terms were also tested but excluded because of the limited relevance of the resulting studies: It's better to share few words/phrases that considered but were excluded. This will help the reader.

Results

There is some inconsistency in the numbers presented. In the PRISMA chart (41628-41377), it should be 251. But chart shows 252. The full text screening section needs to list the articles that were included and how many articles were excluded. Also, i.e. exclusion should be 211 articles (251-211=40) but not clear. The number of studies is 39 in the text above, but the table lists 40 studies.

Also, Out of 40 included 31 were primary articles. The details regarding remaining 9 articles can be provided. The data extraction pattern is separate for the review paper and policy documents. Details are needed on how the findings from the commentaries, reviews, policy documents, and letters differed from the original articles

Types of Literature: Out of all reviewed articles, 70% were quantitative (n = 28), 5 % were qualitative (n = 2), and ~3 % (n=1) used mixed methods. Also included were five commentaries (13%), one systematic literature review (3%), one letter (3%), and one policy document (3%).

Discussion

Expand the abbreviation on first use IAPT

Reviewer #2: Thank you for the opportunity to review this interesting manuscript. I have a few concerns with this paper.

1. Firstly, the manuscript is not suited to a public health journal in its present form, as it is largely raising questions around ethics and politics, and the acceptability of lockdowns, rather than the implications for global health ethics or health access or the impact of the lockdown or perceptions around lockdowns or distrust on health access or the functioning of the health system. My suggestion for the authors would be to include aspects of either global health access or ethics, to make it more relevant for publication in the journal.

2. Secondly, and relatedly, I am concerned about the limited generalizability of the implications of this study. Since lockdowns are a once in a lifetime event as a global pandemic of this scale is unlikely to be frequent, the authors are unable to convey the meaning and implications of the findings to either governments, global development agencies or health systems. What do we take away from this analysis that can enable greater systemic preparedness in the future? And is that preparedness one for governments or systems?

3. The themes investigated in this review largely focus on trust, pressure and coercion, but these are political constructs rather than investigated in the context of health. How do the authors justify the relevance of this paper to the journal?

4. “The review identified major gaps in our knowledge pertaining to the absence of information regarding specific individual characteristics and circumstances that increase the likelihood of experiencing perceived coercion and its related constructs. It also highlighted the absence of standardised measures of perceived coercion, pressures and procedural justice that could be adopted globally, and a need for a better understanding of the cultural and socioeconomic factors affecting adherence to lockdown.” My suggestion for the authors is to provide greater clarity in the abstract as to what those gaps are instead of referring to them. Are the measures needed for health systems and where or in what frameworks would these measures be used?

6. PLOS authors have the option to publish the peer review history of their article (what does this mean?). If published, this will include your full peer review and any attached files.

**Do you want your identity to be public for this peer review?** For information about this choice, including consent withdrawal, please see our Privacy Policy.

Reviewer #1: No

Reviewer #2: No

---

## [Decision Letter · Decision Letter 1]

18 Jan 2023

PGPH-D-22-01026R1

Perceived coercion, perceived pressures and procedural justice arising from global lockdowns during the COVID-19 pandemic: a scoping review

Dear Dr.Ranieri,

Thank you for submitting your manuscript to PLOS Global Public Health. After careful consideration, we feel that it has merit but does not fully meet PLOS Global Public Health’s publication criteria as it currently stands. Therefore, we invite you to submit a revised version of the manuscript that addresses the points raised during the review process.

I have reviewed the comments of the reviewers and I would like to draw your attention to the detailed and useful suggestions for revising your manuscript that have been given. I feel the points raised by Reviewer 4 have merit and hence would like to draw your attention to their comments.

If you could address these issues, then we will be able to take a final call on the suitability for publication of your manuscript in PLOSGPH. 

I would also like to apologise for the delay in turning this around. This was due to the difficulty in obtaining reviewers for your submission. 

We look forward to receiving your revised manuscript.

Kind regards,

Mathew Sunil George

Academic Editor

Journal Requirements:

Additional Editor Comments (if provided):

Reviewers' comments:

Reviewer's Responses to Questions

**Comments to the Author**

1. If the authors have adequately addressed your comments raised in a previous round of review and you feel that this manuscript is now acceptable for publication, you may indicate that here to bypass the “Comments to the Author” section, enter your conflict of interest statement in the “Confidential to Editor” section, and submit your "Accept" recommendation.

Reviewer #3: All comments have been addressed

Reviewer #4: All comments have been addressed

2. Does this manuscript meet PLOS Global Public Health’s publication criteria? Is the manuscript technically sound, and do the data support the conclusions? The manuscript must describe methodologically and ethically rigorous research with conclusions that are appropriately drawn based on the data presented.

Reviewer #3: Yes

Reviewer #4: Yes

3. Has the statistical analysis been performed appropriately and rigorously?

Reviewer #3: Yes

Reviewer #4: N/A

4. Have the authors made all data underlying the findings in their manuscript fully available (please refer to the Data Availability Statement at the start of the manuscript PDF file)?

Reviewer #3: Yes

Reviewer #4: Yes

5. Is the manuscript presented in an intelligible fashion and written in standard English?

Reviewer #3: Yes

Reviewer #4: Yes

6. Review Comments to the Author

Reviewer #3: The authors have done a good job of addressing all the reviews

Reviewer #4: Report attached.

For soem, reason when I pasted the review it said "Minimum character count not met" even though the review was quite long! So I am attaching the review as a separate document and uploading it.

7. PLOS authors have the option to publish the peer review history of their article (what does this mean?). If published, this will include your full peer review and any attached files.

**Do you want your identity to be public for this peer review?** For information about this choice, including consent withdrawal, please see our Privacy Policy.

Reviewer #3: **Yes: **Madhukar Pai

Reviewer #4: **Yes: **Prashanth N Srinivas

---

## [Decision Letter · Decision Letter 2]

10 Feb 2023

Perceived coercion, perceived pressures and procedural justice arising from global lockdowns during the COVID-19 pandemic: a scoping review

PGPH-D-22-01026R2

Dear Dr Veronica Ranieri,

We are pleased to inform you that your manuscript 'Perceived coercion, perceived pressures and procedural justice arising from global lockdowns during the COVID-19 pandemic: a scoping review' has been provisionally accepted for publication in PLOS Global Public Health.

Best regards,

Muhammed O Afolabi, MD, MPH, PhD

Academic Editor

Reviewer Comments (if any, and for reference):

Reviewer's Responses to Questions

**Comments to the Author**

1. If the authors have adequately addressed your comments raised in a previous round of review and you feel that this manuscript is now acceptable for publication, you may indicate that here to bypass the “Comments to the Author” section, enter your conflict of interest statement in the “Confidential to Editor” section, and submit your "Accept" recommendation.

Reviewer #3: All comments have been addressed

Reviewer #4: All comments have been addressed

2. Does this manuscript meet PLOS Global Public Health’s publication criteria? Is the manuscript technically sound, and do the data support the conclusions? The manuscript must describe methodologically and ethically rigorous research with conclusions that are appropriately drawn based on the data presented.

Reviewer #3: Yes

Reviewer #4: Yes

3. Has the statistical analysis been performed appropriately and rigorously?

Reviewer #3: Yes

Reviewer #4: N/A

4. Have the authors made all data underlying the findings in their manuscript fully available (please refer to the Data Availability Statement at the start of the manuscript PDF file)?

Reviewer #3: Yes

Reviewer #4: Yes

5. Is the manuscript presented in an intelligible fashion and written in standard English?

Reviewer #3: Yes

Reviewer #4: Yes

6. Review Comments to the Author

Reviewer #3: The authors have addressed two rounds of peer review and have done their best to address the feedback

Reviewer #4: Thank you for the opportunity to review the revised version of this article. I compliment the authors on a very careful revision that substantively addresses the issues raised in my review report. I still have minor "disagreements" with the framing of the Discussion (see my points raised in the review). The authors have shared their view on this and although I disagree with them, this is outside the scope of a peer review.

7. PLOS authors have the option to publish the peer review history of their article (what does this mean?). If published, this will include your full peer review and any attached files.

**Do you want your identity to be public for this peer review?** For information about this choice, including consent withdrawal, please see our Privacy Policy.

Reviewer #3: **Yes: **Madhukar Pai

Reviewer #4: **Yes: **Prashanth N Srinivas
